# Ciprofloxacin-Loaded Mixed Polymeric Micelles as Antibiofilm Agents

**DOI:** 10.3390/pharmaceutics15041147

**Published:** 2023-04-04

**Authors:** Rumena Stancheva, Tsvetelina Paunova-Krasteva, Tanya Topouzova-Hristova, Stoyanka Stoitsova, Petar Petrov, Emi Haladjova

**Affiliations:** 1Institute of Polymers, Bulgarian Academy of Sciences, Akad. G. Bonchev St. bl. 103-A, 1113 Sofia, Bulgaria; rstancheva@polymer.bas.bg (R.S.); ppetrov@polymer.bas.bg (P.P.); 2The Stephan Angeloff Institute of Microbiology, Bulgarian Academy of Sciences, Akad. G. Bonchev St. bl. 26, 1113 Sofia, Bulgaria; pauny@abv.bg (T.P.-K.); stoitsova_microbiobas@yahoo.com (S.S.); 3Faculty of Biology, Sofia University “St. K. Ohridski”, 8 D. Tsankov Blvd., 1164 Sofia, Bulgaria; topouzova@biofac.uni-sofia.bg

**Keywords:** polymer micelles, co-assembly, drug carriers, bacterial biofilm, biocompatibility

## Abstract

In this work, mixed polymeric micelles (MPMs) based on a cationic poly(2-(dimethylamino)ethyl methacrylate)-b-poly(ε-caprolactone)-b-poly(2-(dimethylamino)ethyl methacrylate) (PDMAEMA_29_-b-PCL_70_-b-PDMAEMA_29_) and a non-ionic poly(ethylene oxide)–b-poly(propylene oxide)–b-poly(ethylene oxide) (PEO_99_-b-PPO_67_-b-PEO_99_) triblock copolymers, blended at different molar ratios, were developed. The key physicochemical parameters of MPMs, including size, size distribution, and critical micellar concentration (CMC), were evaluated. The resulting MPMs are nanoscopic with a hydrodynamic diameter of around 35 nm, and the ζ-potential and CMC values strongly depend on the MPM’s composition. Ciprofloxacin (CF) was solubilized by the micelles via hydrophobic interaction with the micellar core and electrostatic interaction between the polycationic blocks, and the drug localized it, to some extent, in the micellar corona. The effect of a polymer-to-drug mass ratio on the drug-loading content (DLC) and encapsulation efficiency (EE) of MPMs was assessed. MPMs prepared at a polymer-to-drug mass ratio of 10:1 exhibited very high EE and a prolonged release profile. All micellar systems demonstrated their capability to detach pre-formed Gram-positive and Gram-negative bacterial biofilms and significantly reduced their biomass. The metabolic activity of the biofilm was strongly suppressed by the CF-loaded MPMs indicating the successful drug delivery and release. The cytotoxicity of empty and CF-loaded MPMs was evaluated. The test reveals composition-dependent cell viability without cell destruction or morphological signs of cell death.

## 1. Introduction

Bacterial infections are a serious threat to human health. A lot of them are related to the formation of bacterial biofilms that represent sessile communities of bacterial cells held together by extracellular polymeric substances [1,2,3]. The complex structure of bacterial biofilms limits the effective penetration of active substances into bacterial cells [3,4]. The structural characteristics of biofilms create a variety of self-protective mechanisms that result in tolerance to antimicrobial drugs [5,6]. Hence, a large proportion of biofilm-related infections are due to the colonization of human tissues or medical implants [7,8]. Biofilm-related health hazards can also be a result of the contamination of hospital environment surfaces, [5] water pipelines, [9,10,11] foods and the surfaces in food-processing facilities, [12], etc. The intrinsic tolerance of biofilms to drugs or conventional disinfectants requires the development of novel approaches to biofilm eradication. One prospective direction is the application of nanomaterials as both biofilm-destruction agents and drug carriers [13,14,15,16,17,18].

Ciprofloxacin (CF) is a wide-spectrum antibiotic approved by the FDA against various bacterial infections [5,6]. Effective antimicrobial therapy, however, depends on CF solubility and its efficient delivery to the target site of infection [19,20].

Polymeric micelles (PMs) have been extensively studied as drug delivery carriers [21,22,23,24,25]. They represent colloidal particles usually in the nanoscale range (10–200 nm) formed by the self-assembly of amphiphilic block copolymers in aqueous media. PMs are characterized by a core–shell structure, which is directly related to their beneficial properties. The hydrophobic core successfully solubilizes poorly soluble active substances, while the hydrophilic shell protects molecules and prolonged blood circulation time. The advantages of PMs include large encapsulation efficiency, high bioavailability, as well as controlled and targeted drug release. In addition, PMs are characterized by good thermodynamic and kinetic stability, allowing improved pharmacokinetics and biodistribution of antibiotics, enhanced antibacterial efficacy against biofilm-related infectious diseases and capability for overcoming cellular and tissue barriers. Used for antibacterial purposes, PM-based systems have been effectively loaded with antibiotics, providing their homogeneous diffusion in the biofilm matrix [26]. Besides the improved solubility and prevented overdose of antibiotics, PMs carrying a positive charge have been found to exhibit strong antibacterial activity themselves [27,28,29]. This is due to the large surface area and high charge density of the micellar shell that easily interacts with microbial membranes. The latter causes membrane destabilization and successful penetration of PMs resulting in the growth inhibition or killing of the bacteria. An important advantage is that PM-based antimicrobial agents do not induce bacterial resistance [30,31]. Cationic PMs usually carry quaternary ammonium or tertiary amino moieties [27,28,29]. Such micellar systems based on chitosan, [32,33] poly[2-(*tert*-butylaminoethyl) methacrylate], [34] poly(amidoamine), [35,36] poly[2-(dimethylamino)ethylmethacrylate, [37,38] poly(β-amino ester) [39], etc., have been found to exhibit good antibacterial efficacy against wide-spread pathogenic bacterial strains. However, the polycations are usually associated with pronounced cytotoxicity, especially when administered in higher doses [40,41].

The preparation of mixed polymeric micelles has been a successful approach for improving and optimizing the useful properties of PMs [42,43]. The co-assembly of two or more dissimilar copolymers provides the formation of robust systems, having all the advantages of the building components [42,43]. The use of non-ionic segments such as polyoxyethylene to build a part of the hydrophilic MPMs corona led to improved biocompatibility [42,43,44]. Therefore, the preparation of MPMs bearing both cationic and non-ionic moieties might be a good alternative for developing novel biocompatible anti-biofilm agents.

Recently, we have shown the potential of cationic PMs based on PDMAEMA-b-PCL-b-PDMAEMA triblock copolymers for the dispersal of bacterial biofilms [45,46]. The single micellar systems have shown a good reduction of biofilm biomass and vitality of the bacteria in the biofilms, although a slightly increased toxicity was observed. Considering their promising behavior as antibiofilm agents, in this work, MPMs based on PDMAEMA_29_-b-PCL_70_-b-PDMAEMA and the non-ionic PEO_99_-b-PPO_67_-b-PEO_99_ triblock copolymer (known as Pluronic F127) were prepared. Pluronic F127 was selected because these copolymers are approved by FDA and considered biocompatible and safe for clinical trials [47]. The MPMs were formed at different copolymer molar ratios and characterized by dynamic and electrophoretic light scattering. To strengthen their antibacterial properties, MPMs were loaded with the antibiotic CF. The encapsulation efficiency, drug-loading content, and drug release profile of the elaborated system were determined. The cytotoxicity of the resulting drug delivery systems was evaluated as well. The biomass reduction of pre-formed bacterial biofilms and their metabolic activity were estimated.

## 2. Materials and Methods

### 2.1. Materials

All reagents and solvents were of analytical grade, purchased from Sigma-Aldrich (St. Louis, MO, USA), and used without any further purification. Ciprofloxacine (98%, Acros Organics, Geel, Belgium) was purchased from Fisher Scientific (Waltham, MA, USA). Pluronic F127 (M_w_ = 12,500 g·mol^−1^) was kindly provided by BASF Corporation, Ludwigshafen am Rhein, Germany. For the preparation of micellar dispersions, ultra-pure water (>18 MΩ) was used.

#### 2.1.1. Cationic Triblock Copolymer

The amphiphilic PDMAEMA-b-PCL-b-PDMAEMA triblock copolymer was synthesized according to procedures described elsewhere [48]. Briefly, PDMAEMA_29_-b-PCL_70_-b-PDMAEMA_29_ triblock copolymer (M_n_ = 17,100 g·mol^−1^, Đ = 1.20, polycationic content of 53 wt%) was obtained by ATRP of DMAEMA initiated by a bifunctional PCL macroinitiator (M_n_ = 8000 g·mol^−1^, Đ = 1.52).

#### 2.1.2. Preparation of Polymeric Micelles

Single-component PMs (SCPMs) were prepared by dissolving an appropriate amount of the block copolymer (Pluronic F127 or PDMAEMA-b-PCL-b-PDMAEMA) in tetrahydrofuran, followed by dropwise addition of the organic solution to deionized water. MPMs were prepared by dissolving appropriate amounts of the 2 block copolymers at molar ratios 3:1, 1:1, and 1:3 in tetrahydrofuran. Then, the blended copolymer solution was added dropwise to deionized water. Afterwards, all dispersions were subjected to dialysis against water for 5 days through a dialysis membrane (SpectraPore 7, MWCO 8000) to remove the organic solvent. The final concentrations of all micellar dispersions were 1 mg·mL^−1^.

#### 2.1.3. Loading of Polymeric Micelles with Ciprofloxacin

Loading SCPMs and MPMs with CF was performed by addition of drug powder to the micellar dispersion (1 mg·mL^−1^) at polymer-to-drug mass ratio in the range of 1:1 to 50:1. The mixtures were sonicated for 1 h at 60 °C. The dispersions were filtered through sterile PES membrane filters with pore size of 0.2 μm to collect the insoluble CF. The filters were then rinsed with methanol to dissolve CF, and the non-loaded (free CF) fractions were quantified chromatographically and spectrophotometrically. The encapsulation efficiency (EE) and drug loading content (DLC) were determined by the following equations:$$\mathrm{EE}\left(\%\right)=\frac{\mathrm{total}\;\mathrm{amount}\;\mathrm{CF}\;\mathrm{added}-\mathrm{free}\;\mathrm{amount}\;\mathrm{CF}}{\mathrm{total}\;\mathrm{amount}\;\mathrm{CF}\;\mathrm{added}}*100$$
$$\mathrm{DLC}\;\left(\%\right)=\frac{\mathrm{entrapped}\;\mathrm{CF}\;\mathrm{amount}}{\mathrm{micellar}\;\mathrm{weight}}*100$$

#### 2.1.4. In Vitro Release of CF

The release of CF was performed in phosphate buffer (pH 7.4). The CF-loaded PMs (total volume of 3 mL) were placed in dialysis membrane (SpectraPore 7, MWCO 50,000), and the membrane was immersed in 30 mL of the dissolution medium at 37 °C. Aliquots of samples were taken from the dissolution medium at specific time intervals, and that volume was replaced with fresh medium to maintain sink conditions. The amount of released CF was determined by high-performance liquid chromatography (HPLC) and UV-VIS spectrophotometry.

### 2.2. Methods

#### 2.2.1. Dynamic and Electrophoretic Light Scattering

Dynamic and electrophoretic light scattering was performed on a NanoBrook 90Plus PALS instrument (Brookhaven Instruments Corporation, Holtsville, NY, USA) equipped with a 35 mW red diode laser (λ = 640 nm). DLS measurements were carried out at a scattering angle (θ) of 90° and temperature of 25 °C. The hydrodynamic diameter, D_h_, was calculated using the equation of Stokes–Einstein. The ζ-potential measurements were carried out at a scattering angle (θ) of 15° using ELS and PALS methods. The ζ-potentials were calculated from the obtained electrophoretic mobility at 25 °C by the Smoluchowski equation.

#### 2.2.2. Determination of Critical Micellar Concentration

CMCs values were determined by the dye solubilization method using the hydrophobic dye 1,6-diphenyl-1,3,5-hexatriene (DPH). Micellar dispersions of 2 mL at 10 different concentrations in the 0.001–0.5 mg·mL^−1^ range were prepared by diluting a stock solution. Then, 20 μL of DPH solution in methanol (0.4 mM) was added to each sample. The samples were incubated in the dark at room temperature. After 18 h, UV–VIS absorption spectra of DPH were recorded in the λ = 300–500 nm range. The CMC of each sample was determined from the break of the absorbance intensity at 356 nm vs. concentration curve.

#### 2.2.3. High-Performance Liquid Chromatography

HPLC was used to analyze buffer and methanol probes with Avantor ACE 5 C18 column (P/N:ACE-121-1204-125 mm length and 4.0 mm ID) at 1.0 mL.min^−1^ flow rate. A Shimadzu Nexera 40XR HPLC system composed of the following components was used for performing the gradient elution for 15 min: LC40 solvent delivery up to 700 bar pressure; SIL 40C XR automatic sampler with cooling system; CTO-40S column oven up to 100 °C; and SPD-M40 photo diode array detector-scan range 190–800 nm. The mobile phase A contains methanol/buffer solution (25 mM *o*-phosphoric acid with pH 3.0 and 1 mL Et3N) = 10:90 *v*/*v*. Mobile phase B contains methanol/buffer solution (25 mM *o*-phosphoric acid with pH 3.0 and 1 mL Et3N) = 40:60 *v*/*v*. All solutions were filtered through 0.2-µm RC-membrane filters before injection, including the mobile phases. A 2 µL sample was automatically injected for analysis.

#### 2.2.4. UV–VIS Spectrophotometry

All UV–VIS spectra were taken on a Beckman Coulter DU^®^ 800 spectrophotometer by using quartz cell with a path length of 1 cm.

#### 2.2.5. Biofilm Experiments

The strains used in this study were *Escherichia coli* ATCC 25922 and *Staphylococcus aureus* ATCC 29213. The bacteria were kept as stocks at −80 °C with 8% DMSO. Short-term maintenance was performed on slant Nutrient Agar (HiMedia, Bedford, PA, USA) for *E. coli* or Trypticase Soya Agar (HiMedia) for *S. aureus*. Overnight, 18-h cultures of the strains at 37 °C in the respective liquid media were used as a source of inoculum for biofilm cultivation.These cultures were diluted 1:100 in M63 minimal salt medium (0.02 M KH_2_PO_4_, 0.04 M K_2_HPO_4_, 0.02 M (NH_4_)_2_SO_4_, 0.1 mMMgSO_4,_ and 0.04 M glucose, pH 7.5). The bacterial suspensions were loaded on 96-well, U-bottomed polystyrene microtiter plates (Corning, Corning, NY, USA), 150 µL per well, and biofilms were cultivated for 24 h at 37 °C under static conditions. Then, the plankton was discarded, the wells were washed with 3 changes of PBS, and the micelles (1 mg·mL^−1^ of aqueous solution) were added to the washed biofilm. Each micelle was applied in 6 repeats. To check the starting amount of the biofilm, 2 6-well control lanes were included—1 filled with deionized sterile H_2_O (i.e., the solvent included in the micellar suspensions) and 1 left with no addition of liquid, just the biofilm at time point “0” of the treatment trial. A separate lane was parallelly treated with 70 mg/mL of CF, i.e., the biggest amount of the antibiotic applied with the micelles. Incubation with the micelles was performed for 4 h at 37 °C, the micelles were removed, and the biofilm was colored for 15 min with 0.1% of aqueous crystal violet. Then, the wells were thoroughly washed with PBS, and the samples with *E. coli* were solubilized with 70% ethanol and those with *S. aureus*—with a dilution of 95% ethanol and acetone (4:1). The absorbance was measured at 570 nm wavelength using a plate reader. Quantitative data were analyzed using Excell and ANOVA. Since there was no statistically significant difference between the two controls—filled with dH_2_O and no additional liquid, the values were pooled (further denominated as “control”). All experimental data were calculated as percentage of the average control value and are represented graphically as “% of control_av”.

#### 2.2.6. Biofilm Metabolic Activity

The effect of the micelles on the metabolic activity of the biofilm during treatment was estimated by the reduction of the dye resazurin (the commercial preparation Alamar blue—Invitrogen, Waltham, MA, USA, was used). The principle of the test is that resazurin (blue dye), the active ingredient of Alamar blue, is reduced by living cells to resorufin (pink). The marker was used as recommended by the producer with some modifications [42]. Briefly, the biofilm and the treatments were performed as above, but 5 µL of the dye was added to each well simultaneously with the micelles. The plates were shaken briefly, and the biofilms were incubated for a 24 h period. On time points 4 h and 24 h, the absorbance was measured at 570 and 620 nm. Then, the percentage of reduced dye was calculated, see [42], using the ε_ox_ and ε_red_ values provided by the producer. In this series of experiments, a control containing M63 medium was included.

#### 2.2.7. Cytotoxicity of the Micelles

The cytotoxicity of MPMs was estimated by the MTT test performed on human diploid fibroblasts (HSF). The cells were cultivated for 24 h on 96-well flat-bottomed plate at a starting concentration of 2 × 10^4^ cells per well. The treatment was performed for 4 h with micelles prepared ex tempore as 1 mg·mL^−1^ in RPMI medium without serum and antibiotic–antimycotic supplementation. Controls comprised cells without treatment or treated with 70 mg/mL of CF. The test followed the provider’s (ApliChem, Council Bluff, IA, USA) protocol. After the treatment, the micelles were discarded, the wells were washed with sterile PBS, and the MTT reagent was added. Incubation was performed in the dark until the release of the violet formazan reaction product. After 2 h, the formazan crystals were solubilized with stop-medium (10% SDS and 0.01% HCl in dH_2_O) for 12 h at 37 °C. The absorbance was measured at 570 nm by Epoch Microplate.

Spectrophotometer with the Gen5 Data Analysis software was used. The data are presented as percentage of the average control value (the absorbance of untreated cells). Cell morphology was evaluated by crystal violet staining of a duplicated 96-well plate and observation under inverted phase-contrast microscope at 100 and 200× magnification.

## 3. Results and Discussion

### 3.1. Preparation of Mixed Polymeric Micelles

MPMs were prepared from a cationic PDMAEMA-b-PCL-b-PDMAEMA (hereafter for simplicity noted as PDMAEMA) and a non-ionic PEO-b-PPO-b-PEO (hereafter noted as F127) triblock copolymers at 3 molar ratios (3:1, 1:1, and 1:3). The schematic illustration of MPMs formation is presented on Figure 1. For comparison, SCPMs were formed as well. All micellar systems were prepared by the solvent exchange dialysis method at a concentration of 1 mg·mL^−1^.

The amphiphilic nature of the two copolymers as well as the miscibility of the hydrophobic blocks (PCL and PPO) [49] supposes their co-assembly into structures composed of a mixed PCL/PPO core and a mixed PDMAEMA/PEO shell. It is known that self-association occurs above a certain concentration (CMC) that is specific for each polymer [21,22,23,50]. Moreover, CMC is influenced by the hydrophilic–lipophilic balance (HLB) of the copolymers. For the amphiphilic block copolymers used in this study, HLB was calculated according to Griffin’s method (see Appendix A) and was 14 for F127 and 10.6 for PDMAEMA, respectively. The HLB values imply that PDMAEMA is more hydrophobic than F127. The CMC of the present mixed systems was determined by the dye solubilization method. We used the hydrophobic dye DPH, which after solubilization into the hydrophobic micellar core, exhibited a characteristic UV absorbance band in the range of 300 to 500 nm. The CMC values were determined from the break of the absorbance intensity taken at 356 nm vs. copolymer concentration curves (Appendix A). The resulting values are summarized in Table 1. It could be seen that the CMC was dependent on the micellar composition. Expectedly, F127 exhibited higher CMC than PDMAEMA due to the weaker hydrophobicity of PPO than PCL. [51] Increasing the fraction of the more hydrophobic PCL into the micellar core of MPMs decreased their CMC values.

The MPMs were also characterized by electrophoretic and dynamic light scattering. Table 1 lists the determined values of ζ-potential. It is evident that the SCPMs based on the cationic PDMAEMA polymer exhibited strong positive ζ-potential (29.1 mV), while the value of those based on the non-ionic F127 tends to be neutral. As expected, the surface charge of MPMs was dependent on the composition of the mixed micelles. Increasing the molar fraction of F127 resulted in a gradual decrease in the surface charge of MPMs. Thus, the good correlation between the sample compositions and the change of CMC and ζ-potential is an indication of the formation of MPMs from the two block copolymers.

The formation of one population of particles from the PDMAEMA and F127 blend, i.e., of mixed micelles only, was proved by additional experiments. Figure 1 compares the size distribution and ζ-potential of SCPMs formed from PDMAEMA and F127, a physical mixture (1:1 *v*/*v*) of the 2 pre-formed SCPMs, as well as the MPMs at 3 different compositions. All samples, except the physical mixture, possessed monomodal ζ-potential and size distribution curves. The presence of two different populations of micelles was registered for the physical mixture with peak maximum values identical to those of the SCPMs (Figure 1a,c). The fact that pre-formed SCPMs, formed from PDMAEMA and F127, did not merge after mixing can be attributed to the superior structural stability of the micelles comprising a PCL core. Due to the semi-crystalline nature of PCL, such aggregates are kinetically frozen (non-dynamic micelles) and disintegrate at a very slow rate [52]. Therefore, in the physical mixture, the unimers exchange between PDMAEMA and F127 micelles was not pronounced, and no structural reorganization towards the formation of mixed aggregates was observed. The stabilizing role of PCL was further demonstrated by a comparative experiment where the micellar solutions were diluted to a concentration which is below their CMC value (0.01 mg·mL^−1^). The presence of MPMs in the samples was registered, and their size distribution curves, correlation curves, and phase plots remained unaffected by the dilution (Appendix A). In contrast, the shape of the correlation curve and phase plot for the dynamic F127 micelles dramatically changed, which is attributed to the transition from aggregates to unimers (Appendix A). Taken together, the results from DLS and ELS experiments undoubtedly revealed that blending the two copolymers in solution and the subsequent replacement of the organic solvent with water provides well-defined mixed micelles.

The size of MPMs was not influenced by their composition. As evident from Table 1, the hydrodynamic diameter, D_h_, of mixed particles vary from 34.6 to 37.1 nm. D_h_ values of MPMs were closer to the D_h_ of PDMAEMA SCPMs (33.2 nm) rather than the D_h_ of F127 micelles (14.9 nm). Obviously, the more hydrophobic PDMAEMA triblock copolymer has a leading role in the process of micelle formation and, to some extent, determines their dimensions.

### 3.2. Loading of Micelles with CF

CF is a lipophilic fluoroquinolone antibiotic with a zwitterionic character and limited solubility (<0.3 mg·mL^−1^) in aqueous media at physiological pH [19,20]. Therefore, to enhance the solubility of CF, we performed a series of experiments focused on loading CF in MPMs at various polymer-to-drug mass ratios, ranging from 1:1 to 50:1. The mixtures were first sonicated for 1 h at 60 °C to ensure maximum drug solubilization and encapsulation, and then filtered to collect the insoluble drug fraction. The filtered aqueous dispersions were analyzed by dynamic and electrophoretic light scattering, while the filter fractions were used to determine the encapsulation efficiency and drug-loading content.

The EE and DLC of all micellar systems were determined spectrophotometrically from the characteristic absorbance band of CF at 280 nm. It could be seen from Figure 2 that the two parameters depended on the polymer-to-drug mass ratio and the micellar composition. At low CF contents (50:1 and 10:1), the EE was very high (90–98%, Figure 2a) and then tended to decrease. At high CF contents (2.5:1 and 1:1), the EE of F127 SCPMs was remarkably lower than the EE of the other compositions. In a similar fashion, the DLC of PMs was influenced by the micellar composition and polymer-to-drug mass ratio. Again, at the higher CF contents (2.5:1 and 1:1), SCPMs formed from F127 exhibited lower DLC values. In contrast, the PDMAEMA SCPMs and MPMs showed relatively high loading ability. The obtained DLC values were in good agreement with the results reported for similar drug delivery systems [53,54,55].

In order to validate the loading capacity of the investigated micellar systems, the EE and DLC were determined by HPLC. The CF was detected at 280 nm and was eluted at 9.01 to 9.2 min. A single symmetric peak was observed at all samples with a maximum corresponding to those of pure CF used as a standard. Representative chromatograms are available in the Appendix A. The determined values of EE and DLC by both UV–Vis spectrophotometry and HPLC were in good correlation (Appendix A), except for the samples prepared at 2.5:1 and 1:1 polymer-to-drug mass ratios. It should be mentioned that EE determined for these two ratios were rather low, and therefore, these two systems were excluded from our further experiments.

The variations of D_h_ and ζ-potential of CF-loaded micelles are presented in Figure 3. It is noticed that the addition of CF did not significantly influence either the size or the ζ-potential of SCPMs (Figure 3a,e). In contrast, embedding CF into MPMs evoked some increase in their size and reduced their ζ-potential. In particular, the D_h_ of MPMs remained unchanged up to 10:1 and started to increase at a 5:1 polymer-to-drug mass ratio. The ζ-potential, however, was highly influenced by the CF concentration, decreasing linearly with the rise of the drug. These findings could be attributed to the zwitterionic character of CF at neutral pH, as its ionization degree increases with decreasing the pH [56]. On the other hand, PDMAMEA is a weak polybase, and, in an aqueous medium, an electrostatic interaction between the polycation and CF could take place [57,58]. As a result, the positively charged PDMAEMA chains became partially neutralized, indicated by a drop of ζ-potential. The change of ζ-potential was in correlation with the MPMs composition, in particular with the molar fraction of PDMAEMA in the micellar shell (3:1, 1:1, and 1:3). The increased D_h_ of MPMs at a 5:1 polymer-to-drug mass ratio could also be attributed to the neutralization of PDMAEMA chains that provoke particle aggregation. It should be mentioned that pronounced electrostatic interaction between CF and PDMAEMA SCPMs was not detected from dynamic and electrophoretic light scattering. This is probably due to the fact that at those micelles the shell is composed only of PDMAEMA, and a large amount of CF is required to trigger a change in D_h_ and ζ-potential.

Based on the physicochemical characterization of the investigated micellar systems, we choose to continue our further experiments with PMs obtained at a 10:1 polymer-to-drug mass ratio as an optimal system.

### 3.3. CF In Vitro Release

CF release from the present micellar systems was investigated at a physiological temperature (37 °C) in a phosphate buffer at pH 7.4, resembling an extracellular fluid pH. The amount of released drug was determined by both HPLC and UV–Vis spectrophotometry. The characteristic UV–Vis spectra and HPLC chromatograms are available in the Appendix A. CF release profiles of different PMs, determined by HPLC, are shown in Figure 4. As can be seen, the profiles are characterized by an initial fast release, followed by a sustained release up to the end of the investigated time period. The amount of released drug in the first 4 h was from about 50 to 80%, depending on the micellar composition. Generally, the micelles containing more F127 (SCPMs and 1:3 MPMs) exhibited a lower CF release rate, while those comprising more PDMAEMA (PDMAEMA SCPMs, 3:1 and 1:1 MPMs) released the drug more quickly. This was associated with the CF molecules located in the micellar corona, as well as with the sensitivity of PDMAEMA chains to changes in temperature and pH [42]. In other words, under the release conditions (pH 7.4 and 37 °C), the deprotonation of PDMAEMA chains causes the repulsion of CF molecules from the complex. After the initial phase, all profiles exhibited prolonged release, thus demonstrating the potential of MPMs to overcome the side toxic effects related to high dosage and frequency of administration. The same behavior was observed when the samples were analyzed by UV–Vis spectrophotometry (Appendix A).

### 3.4. Effects of the Micelles on the Biofilm Biomass and Metabolic Activity

The ability of the investigated micellar systems to reduce the biofilm biomass was estimated by the crystal violet assay. The two bacterial strains used for the experiments, *E. coli* 25922 and *S. aureus* 29213, are the recommended reference strains for antibacterial susceptibility testing. PDMAEMA-based micellar systems have been previously shown by us to exhibit an exfoliation capacity [45,46]. Therefore, based on our data, we performed tests for biofilm destruction for 4 h. As evident from Figure 5, a stronger effect against the mature biofilm of the Gram-negative model strain, *E. coli* (Figure 5a), was noticed. After the treatment period, a slight influence of F127 SCPMs was observed since only c.a. 20% of the biofilm was removed (Figure 5a). In contrast, all micellar compositions containing PDMAEMA showed a significant reduction (c.a. 70%) of the pre-formed biofilms. ANOVA analysis for this strain showed no significant differences between the different MPMs as well as between MPMs and PDMAEMA SCPMs.

The exfoliating effect of all studied PMs on the Gram-positive model strain, *S. aureus,* biofilms, as seen in Figure 5b, was lower compared to that on *E. coli*. This could be due to the thicker cell envelope characteristic of Gram-positive bacteria, which makes them more resistant than Gram-negative ones [59]. However, an effect of the micellar composition was observed for this strain. By increasing the PDMEMA amount in the micellar corona, the ability of MPMs to reduce the biofilm biomass increased. Unexpectedly, PDMAEMA SCPMs deviated from this behavior as their exfoliating effect was lower than those of MPMs and close to that of F127. It is known that *S. aureus* has increased resistance to cationic compounds due to the lysinylation of phosphatidylglycerol and d-alanylation of the teichoic acids embedded in the inner bacterial membrane [60]. This probably restricts the efficiency of PDMAEMA SCPMs. However, when PEO is present in the micellar corona, a shielding effect is presumably provoked. Moreover, it has been reported that Pluronic micelles could alter various cellular functions, such as mitochondrial respiration, apoptotic signal transduction, the activity of drug efflux transporters, gene expression, etc. [61,62]. Therefore, we could assume that the presence of PEO in the micellar corona enhanced the penetration of MPMs into the *S. aureus* bacterial biofilm. Here, we have to take into account the influence of the positive charge of MPMs that contributed to better interaction with the cellular envelope. Obviously, the presence of PEO in the MPM does not affect notably their zeta potential (see Figure 1 and Figure 2) and strong exfoliating effect, respectively.

Statistical analysis showed that the presence of CF in the micellar composition did not alter their exfoliating activity in comparison with the empty micelles for the two bacterial strains used. CF, when applied alone, had a slight or no effect on the biofilm.

Together with the exfoliating potential, it was important to check whether or not the remaining undetached biofilm was functionally altered and if there was some difference between the redox capacities of sessile bacteria treated with empty to CF-loaded PMs. Therefore, we performed the Alamar blue assay (Figure 6). The reduction of the resazurin dye at 2 time points—4 h and 24 h—were compared. For both strains and all micellar compositions, a change in the dye reduction after 4 h was hardly notable (light grey columns). This is quite reasonable because the biofilm cells are small and form a thin layer. Such a result was in accordance with the producer’s instruction which says that for some cellular types, one may need longer intervals in order to achieve detectable changes in the reagent [63]. After 24 h, however, a detectable increase of the accumulated reduced dye for empty PMs in both the Gram-negative and the Gram-positive model strains was observed (Figure 6, dark grey columns). The difference in these values from the control was likely related to the different amounts of the tested biofilm. When the CF-loaded PMs were used, for 24 h post-treatment, no detectable shift in the resazurin/resorufin proportion was registered. Samples treated with CF alone showed no statistically significant differences from the M63 control. This implies that the PMs managed to deliver the antibiotic to the bacterial cells embedded in the biofilm matrix and protected by it. When referring to the release profiles (Figure 4), we should note that after 4 h, the biofilm bacteria were exposed to the action of about 50% of CF, while after 24 h, the percentage of the released drug was approximately 80. No matter whether these bacteria are killed or are simply dormant, it can be concluded that they have substantially suppressed metabolic activity.

### 3.5. Cytotoxicity and Cellular Morphology

The cytotoxicity of empty and CF-loaded PMs was evaluated by an MTT test performed on human diploid fibroblasts (HSF). As can be seen from Figure 7, all tested PMs exhibited cell viability above 50%. For the empty micelles (light grey columns), a strong composition-dependent cytotoxic effect was observed. As expected, PDMAEMA SCPMs had the strongest effect and suppressed HSF cell viability to c.a. 60%, while the cells quite well tolerated F127 (c.a. 90% viability compared with controls). For the empty MPMs, a trend to increase the HSF tolerance with increasing the PEO amount in the micellar corona was clearly noticed. Thus, the mixed compositions are characterized by c.a. 66, 74, and 86% cell viability for 3:1, 1:1, and 1:3, respectively. This was statistically significant (*p* < 0.01) if compared with the PDMAEMA SCPMs. The PMs loaded with CF exhibited less tolerance to the HSF cells (dark grey columns). Unlike this, the CF, when applied alone, reduced cell viability. It is also noticeable that the effect of the micellar composition was lost for the PDMAEMA-rich compositions (PDMAEMA SCPMs, 3:1 and 1:1 MPs). We can assume that this is probably due to the toxicity of CF itself. This assumption is supported by the drug release tests (Figure 4), where the above-mentioned systems exhibited high release rates.

Additionally, a morphological analysis of human fibroblast (HSF) was performed to analyze the changes in cellular structure (Figure 8). As expected, the compositions that reduced cellular viability (see Figure 7, PDMAEMA SCPMs, 3:1 and 1:1 MPs) showed alterations in the cell structure. Cells appeared stressed, with highly elongated filopodia and a lack of visible lamellipodia, possibly due to disruptions in cell-adhesive contacts and actin cytoskeleton deviations. Increasing the amount of PEO in the micellar corona (compositions 1–3 and F127) significantly reduced these changes, and a distinct fraction of the cell population retained a fibroblast-like morphology. Surprisingly, contrary to the cytotoxicity results (Figure 7), the presence of CF in PMs attenuated the changes in cell morphology. Moreover, the effect of MPMs on the cellular structure was barely distinguishable. It seems that the loading of antibiotics into MPM blocks their effect on the cell-adhesive complexes and thus indirectly reduces the effect on the actin cytoskeleton, which has a major role in the observed morphological changes. These observations were in good agreement with the finding that at the current working conditions, CF interacts electrostatically with PDMAEMA.

## 4. Conclusions

In summary, our study demonstrated that the preparation of MPMs is a feasible approach for preparing effective and biocompatible antibiofilm agents. The properties of MPMs can be easily controlled by the molar ratio of the two blended polymers. The introduction of non-ionic PEO moieties in the micellar shell provided a reduction of the ζ-potential value and hence decreased the cytotoxicity of the systems. PDMAEMA-b-PCL-b-PDMAEMA constituent contributed to the superior structural stability and enhanced loading capacity of MPMs. The favorable physicochemical characteristics and high EE (above 90%) of CF-loaded MPMs obtained at a 10:1 polymer-to-drug mass ratio outlined this system as optimal. The prolonged drug release from MPMs demonstrates the potential of such nanocarriers to overcome problems related to both high dosage and administration frequency. The MPMs were effective in the exfoliation of mature Gram-positive and Gram-negative bacterial biofilms. They suppressed bacterial metabolic activity and were effective in the destruction of biofilms. Due to the controlled CF release, the loaded micelles slightly reduced the viability of HSF cells, however, without cell destruction or morphological signs of cell death. The found dependencies related to the composition of the mixed micelles delineated the system PDMAEMA/F127 3:1 as optimal in terms of biocompatibility and efficiency in the destruction of bacterial biofilms. This system has great potential for application as an antibiofilm agent and can further be developed as an antibiofilm drug carrier of other antibacterial agents such as antibiotics and natural products. Tests on other bacterial model strains as well as on clinical isolates will be performed in the near future.

## Data Availability

Data is contained within this article.

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
