# Peer review of "Ciprofloxacin-Loaded Mixed Polymeric Micelles as Antibiofilm Agents"

_pharmaceutics, 2023, doi:10.3390/pharmaceutics15041147_

Round 1

Reviewer 1 Report

It is interesting that two triblock copolymers, polycation-b-poly(ε-caprolactone)-b-polycation and PEO-b-PPO-b-PEO, form the mixed polymer micelle in aqueous solution. I have two basic questions on this mixed polymer micelle.

(1) Whether is the mixed polymer micelle thermodynamically stable state or formed for only kinetic reason? The following experiment may give an answer to this question. Single component micelles of the two triblock copolymers are mixed in aqueous solution to check the formation of the mixed micelle, for example, by electrophoretic light scattering.

(2) Do the poly(ε-caprolactone) and PPO blocks form a homogeneous phase or phase separate in the mixed micelle core? This question may play an important role in the solubilization of hydrophobic drugs into the mixed polymer micelle. A THF solution of poly(ε-caprolactone) and PPO homopolymers is added dropwise to water, and a DSC measurement may be made on the aqueous dispersion to judge the miscibility of the two homopolymers.

Author Response

Reviewer 1

It is interesting that two triblock copolymers, polycation-b-poly(ε-caprolactone)-b-polycation and PEO-b-PPO-b-PEO, form the mixed polymer micelle in aqueous solution. I have two basic questions on this mixed polymer micelle.

(1)       Whether is the mixed polymer micelle thermodynamically stable state or formed for only kinetic reason? The following experiment may give an answer to this question. Single component micelles of the two triblock copolymers are mixed in aqueous solution to check the formation of the mixed micelle, for example, by electrophoretic light scattering.

Response:

The proposed from the reviewer experiment is a part of Figure 1 (Figure 1a and 1c). We mixed preformed SCPMs from the two copolymers and performed dynamic and electrophoretic light scattering. As mentioned in the text on page 7 “The fact that preformed SCPMs, formed from PDMAEMA and F127, did not merge after mixing can be attributed to the superior structural stability of the micelles comprising a PCL core. Due to the semi-crystalline nature of PCL, such aggregates are kinetically frozen (non-dynamic micelles) and disintegrate with a very slow rate. Therefore, in the physical mixture, the unimers exchange between PDMAEMA and F127 micelles was not pronounced and no structural reorganization towards formation of mixed aggre-gates was observed.”.

To prove our assumptions we performed additional experiment by dynamic and electrophoretic light scattering determining the size and zeta potential of the mixed micelles after dilution below their CMC values. In this regard three new figures were added in the SI (Figure S2a, b and c). It is visible from the size distributions and correlation curves that after dilution mixed micelles and PMMAEMA SCPMs remain in the dispersion. In contrast, the diffusion of F127 SCPMs decrease and the slope of the correlation curve dramatically change due to their desintegration. The similar situation in the phase plots was observed. The later unambiguously demonstrate the kinetic stability of the mixed micelles. An appropriate changes in the text on page 7 were made.

(2) Do the poly(ε-caprolactone) and PPO blocks form a homogeneous phase or phase separate in the mixed micelle core? This question may play an important role in the solubilization of hydrophobic drugs into the mixed polymer micelle. A THF solution of poly(ε-caprolactone) and PPO homopolymers is added dropwise to water, and a DSC measurement may be made on the aqueous dispersion to judge the miscibility of the two homopolymers.

Response:

We fully agree with the reviewer that the formation of a homogeneous hydrophobic phase in the micellar core is important. The PCL and PPO are compatible as a DCS analysis was performed in our previous work cited in the text as ref. 45, after the revision - ref. 49. The thermograms could be find there showing the miscibility of both polymers.

Reviewer 2 Report

Numerous microbial infections are associated with the formation of biofilms, since the bacterial cells organized in this structure are able to better resist disinfectants and antibiotics and attacks by the host's immune system. Ciprofloxacin is an antibiotic belonging to the fluoroquinolone family, used for the treatment of various bacterial infections. To counter them, recent researches concern the application of nanomaterials both as biofilm destruction agents and drug carriers. In this work, mixed polymer micelles loaded with ciprofloxacin as antibiofilm agents were made.

The topic covered in the manuscript is current.

The structure of the paper is well organized:

The introduction informs the reader about the thematic problem and ends with the purpose of the research. The purpose of the article has been correctly defined. The methodology is clearly written and the results and discussion are well presented. Literature selection is appropriate throughout the manuscript.

However, there are some changes to make, as follows:

I think the conclusions section is very important to complete a paper, even if it is not compulsory. However, in this work, this section is too long and reports the results again. Therefore, the authors should summarize the conclusions and suggest future trials.

Author Response

Reviewer 2

Numerous microbial infections are associated with the formation of biofilms, since the bacterial cells organized in this structure are able to better resist disinfectants and antibiotics and attacks by the host's immune system. Ciprofloxacin is an antibiotic belonging to the fluoroquinolone family, used for the treatment of various bacterial infections. To counter them, recent researches concern the application of nanomaterials both as biofilm destruction agents and drug carriers. In this work, mixed polymer micelles loaded with ciprofloxacin as antibiofilm agents were made.

The topic covered in the manuscript is current.

The structure of the paper is well organized:

The introduction informs the reader about the thematic problem and ends with the purpose of the research. The purpose of the article has been correctly defined. The methodology is clearly written and the results and discussion are well presented. Literature selection is appropriate throughout the manuscript.

However, there are some changes to make, as follows:

I think the conclusions section is very important to complete a paper, even if it is not compulsory. However, in this work, this section is too long and reports the results again. Therefore, the authors should summarize the conclusions and suggest future trials.

Response:

The conclusion section was rewritten as the conclusions were summarized and some future trials were suggested.

Reviewer 3 Report

In this work, the authors intend to minimize the cytotoxicity of cationic NPs by inducing non-ionic polymer and meanwhile enhance CIP antibiofilm ability.

1. In the Scheme 1, how to the author ensure the nanocarriers consist of PDMAEMA and F127 polymer. I think some nanoparticle may consist only component.

2. In Figure 5, the author is better to provide the pure CF control. Also in Figure 6.

3. Figure 7 seem peculiar,  in general,  more positive NPs are more toxicity toward HSF cell, but in PMs + CF, it is not, Why?

4. Can author give a best ratio that the synthesized NPs with strong antibiofilm ability and minimizing cytotoxicity.

5. In the introduction, some related references should cited and discussed,  for example, Small, 2022, 19(6):2206220; Nano Today, 2022, 46(1):101602; View, 20202(1):20200065; Journal of Controlled Release, 2021, 329: 1102-1116

Author Response

Reviewer 3

In this work, the authors intend to minimize the cytotoxicity of cationic NPs by inducing non-ionic polymer and meanwhile enhance CIP antibiofilm ability.

  1. In the Scheme 1, how to the author ensure the nanocarriers consist of PDMAEMA and F127 polymer. I think some nanoparticle may consist only component.

Response:

The formation of mixed polymeric micelles due to co-assembly of the two copolymers and is driven by hydrophobic interactions. The formation of mixed structures was evidenced by dynamic and electrophoretic light scattering (Figure 1 and S2) as well as from the determined CMC values (Table 1 and Figure S1). The prerequisites for receiving of mixed micelles without the presence of single component micelles are discussed in details in section 3.1. Preparation of mixed polymeric micelles.

  1. In Figure 5, the author is better to provide the pure CF control. Also in Figure 6.

Response:

The pure CF was added in Figures 5 and 6. Additionally, we provide the CF control in Figure 7. An appropriate changes in the text were made - in the Materials and methods and Results sections, respectively.

  1. Figure 7 seem peculiar, in general, more positive NPs are more toxicity toward HSF cell, but in PMs + CF, it is not, Why?

Response:

The reasons for losing the composition dependence in CF loaded micelles are discussed on page 13, above Figure 7. Generally, this was due to the release of CF from the systems. As evident from the release profiles PDMAEMA SCPMs, 3:1 and 1:1 exhibited high release rates. Therefore, at these systems we have a dominant effect of CF cytotoxicity itself over those of positive charge of the systems.

  1. Can author give a best ratio that the synthesized NPs with strong antibiofilm ability and minimizing cytotoxicity.

Response:

The best ratios regarding the micellar composition and loading with CF are now described in the rewritten conclusion section.

  1. In the introduction, some related references should cited and discussed, for example, Small, 2022, 19(6):2206220; Nano Today, 2022, 46(1):101602; View, 2020, 2(1):20200065; Journal of Controlled Release, 2021, 329: 1102-1116.

Response:

The proposed articles were cited in the text and added to the reference list.

Round 2

Reviewer 3 Report

After revision, the manuscript can be accepted for publication in pharmaceutics.